# Willingness to pay for cataract surgery and associated factors among cataract patients in Outreach Site, North West Ethiopia

**Mohammed Seid[1], Amare Minyihun[2], Gizachew Tilahun[1], Asmamaw Atnafu[2], Getasew Amare[2]***

**1** Department of Optometry, School of Medicine, College of Medicine and Health Science, University of Gondar, Gondar, Ethiopia, **2** Department of Health Systems and Policy, Institute of Public Health, College of Medicine and Health Science, University of Gondar, Gondar, Ethiopia

* getasewa23@gmail.com

## Abstract

### Introduction

In Ethiopia, cataract surgery is mainly provided by donors free of charge through outreach programs. Assessing willingness to pay for patients for cataract surgery will help explain how the service is valued by the beneficiaries and design a domestic source of finance to sustain a program. Although knowledge concerning willingness to pay for cataract surgery is substantive for developing a cost-recovery model, the existed knowledge is limited and not well-addressed. Therefore, the study aimed to assess willingness to pay for cataract surgery and associated factors among cataract patients in Outreach Site, North West Ethiopia.

### Methods

A cross-sectional outreach-based study was conducted on 827 cataract patients selected through a simple random sampling method in Tebebe Gion Specialized Hospital, North West Ethiopia, from 10/11/2018 to 14/11/2018. The data were collected using a contingent valuation elicitation approach to elicit the participants' maximum willingness to pay through face to face questionnaire interviews. The descriptive data were organized and presented using summary statistics, frequency distribution tables, and figures accordingly. Factors assumed to be associate with a willingness to pay were identified using a Tobit regression model with a p-value of <0.05 and confidence interval (CI $\neq$ 0).

### Results

The study involved 827 cataract patients, and their median age was 65years. About 55% of the participants were willing to pay for the surgery. The average amount of money willing to pay was 17.5USD (95% CI; 10.5, 35.00) and It was significantly associated with being still worker (β = 26.66, 95% CI: 13.03, 40.29), being educated (β = 29.16, 95% CI: 2.35, 55.97), free from ocular morbidity (β = 28.48, 95% CI: 1.08, 55.90), duration with the condition, (β = -1.69, 95% CI: -3.32, -0.07), admission laterality (β = 21.21, 95% CI: 3.65, 38.77) and remained visual ability (β = -0.29, 95% CI (-0.55, -0.04).

**Data Availability Statement:** All relevant data are within the paper and its Supporting Information files.

**Funding:** The author(s) received no specific funding for this work.

**Competing interests:** Authors have declared that no competing interests exist.

**Abbreviations:** CV, Contingent Valuation; EDHS, Ethiopian Demographic and Health Survey; ETB, Ethiopian Birr; GDP, Gross Domestic Product; HEP, Health Extension Program; HEW, Health Extension Worker; HP, Health Post; HSDP, Health Sector Development Program; MWTP, Maximum Willingness to Pay.

## Conclusions

Participants' willingness to pay for cataract surgery in outreach Sites is much lower than the surgery's actual cost. Early intervention and developing a cost-recovery model with multi-tiered packages attributed to the neediest people as in retired, less educated, severely disabled is strategic to increase the demand for service uptake and service accessibility.

## Introduction

Globally, cataract is the second leading cause of blindness, accounting for 33% (52.6 million) blind people next uncorrected refractive error [1, 2]. In Ethiopia, 50% of blindness is also caused by cataracts [3]. Cataract surgery is the only treatment modality [4, 5]. According to Vision 2020 initiatives, the targeted cataract surgical rate (CSR) varies globally from 2000–5000 per million [6]. It was planned to be 2000 per million for African countries per year, but now it is less than 500 [7]. This underperformance was reasoned out by low patient demand, poor service delivery [7], low family income [8, 9], unable to afford surgical costs [8, 10, 11], logistical constraints, and fear of the surgery [9, 11], lack of knowledge, lack of family support [11], and time [9]. In 2005, the mean cost of the surgery per eye from various European countries was US$ 843.5 [12]. By the year 2004, in the United States of America, the mean cost was estimated to be US$2525 [13]. Moreover, studies from Africa and South East Asia reported that the actual cost exceeds the community's willingness to pay (WTP) for cataract surgery [14–18]. Overall, the proportion of WTP in Asia was very discrepant, which was ranged from 34% to 90%. Similarly, the mean amount of money participants willing to pay was varied from US$ 7 to US$ 968 [19–22]. However, the mean amount of money participants willing to pay in Africa was ranged from US$2.3 to US$18.5, which was relatively low as compared to Asian nations [19, 21]. In addition to regional variation, WTP for cataract surgery was higher among hospital-based studies and the urban population. However, the findings of outreach-based studies, which were conducted on the rural community, indicated that people are less willing to accept and pay for the surgery. Conclusively, the amount of money WTP is related to setting variation. Moreover, several studies showed that willingness to pay is affected by economic status, gender, age, family support, knowledge about the surgery, preoperative visual ability, family size, occupation, level of perceived need, and the characteristics of local eye care programs even if the majority of analysis model were fitted for the proportion of WTP rather than amount of money to pay [15, 17, 19–21, 23–26]. Besides, WTP is also health state dependant which is affected by the preference and perceived health status of participants. This situation affects the research by over/under estimate result which may not reflect the actual price of the intervention [27–33].

In Ethiopia, it is known that the eye care service is urban-centred, which is not easily accessible for rural dwellers. As a result, cataract surgery is being delivered through outreach programs, with the tremendous aid of donors, either free of charge or at a very low cost. Even if the program is essential and brought Socioeconomic welfare by reducing avoidable blindness associated with cataracts, this might not be financially sustainable for the long-run. Therefore, assessing WTP from the beneficiaries' perspective is very important to understand how the users value the service and design a cost-recovery model that assures self-sustaining and high-volume surgical services [22].

Even if the knowledge regarding willingness to pay and the surgery's actual cost is essential, the existing experience is not going through the intervention. Firstly, the study area's existing

knowledge is mainly centred on the proportion of willingness to pay rather than the amount of money [22]. Secondly, the amount of money WTP was not fully addressed with the appropriate analysis model. Thirdly, willingness to pay is highly setting-variant, which mainly varies with socioeconomic status and study setting. Consequently, it is difficult to apply the research finding done in other corners of the globe for this study area. Therefore, this study aimed to assess the willingness to pay for cataract surgery and associated factors among cataract patients at the Northwest Ethiopia outreach site.

## Methods

### Study design and setting

Outreach- based cross-sectional study was conducted. The campaign site was at Tebebe Gion Specialized Referral Hospital, Bahir Dar city, North West Ethiopia. There are only four Tertiary and general hospitals for about 14 million people in the catchment area, which provide comprehensive eye care services, including cataract surgical services in North West Ethiopia. However, all of them are urban-centred and not accessible for more than 80% of the community. Hence, the outreach program was planned for communities where cataract surgical services are not available and for peoples in disadvantaged areas with low socioeconomic status in North West Ethiopia. Initially, screening for cataracts was conducted in the nearby rural weredas, which are the third-level administrative units in the Ethiopian governance structure, before the campaign. During the screening, Cataract cases were diagnosed and informed about the condition and scheduled to attend the planned outreach program. As a result, a total of 1336 cataract cases were linked and admitted to the campaign from weredas of West Gojjam (628 patients), South Gondar (374 patients), Awi Zone (201 patients), Bahir Dar City (100 patients), and from East Gojjam (33 patients).

Thus, more than 93% of cases were from rural weredas and district towns. Therefore, this study's findings could apply to rural residents admitted for cataract surgery in outreach programs.

### Study population and sampling technique

All adult patients age ≥18 years admitted at the campaign Site, Tebebe Gion specialized Referral Hospital, for cataract surgery was the study population except those who had communication barriers (speech impairment and hearing impairment).

The sample size was determined using single population mean estimation formula, by assuming a 95% confidence level, a standard deviation of WTP from the previous study conducted in Jimma, Southwest Ethiopia (US$7.5 (138.4 ETB)) (24), and a margin of error within 0.5$ around the parameter at 95% confidence level. The final computed sample size, including a 10% non-response rate, was 926.

Thorough sampling procedures were followed to select the study participants. First, screening for cataract was conducted in the nearby weredas so that 1336 cases were appointed for the surgery. The patients were documented and provided identification number. The appointment was based on the rate of surgical procedures planned per day, 170 cataract cases. Thus, the campaign lasted for eight consecutive days. Daily sample participants were calculated proportionally based on the daily patient flow. The sampling frame was designed based on the list of daily admitted patients. Hence, based on this proportion, the study participants were selected from cases prepared for admission using a simple random sampling method.

## Data collection procedures

Data were collected using a structured questionnaire through face to face questionnaire interview from 10/11/2018 to 14/11/2018. The participants' maximum willingness to pay (MWTP) for the cataract surgery interims of monetary value was estimated using a bidding game format of contingent valuation (CV) method [34–38]. The questionnaire was pretested on 47 patients at Ebnat primary hospital, the other equivalent outreach site.

## Measurements and variables

**Maximum willingness to pay (MWTP) for cataract surgery.** Was the independent variable measured by questions prepared with a bidding game eliciting approach. After presenting the hypothetical scenario to assess how participants value the cataract surgery [39], the respondents were asked to accept or reject any positive price for cataract surgery. Those respondents who had accepted to pay some positive fees were asked their MWTP using the bidding format approach. The bidding game's starting price was estimated from the actual cost of *manual small incision cataract surgery* per eye, 2824 ETB from a study done in Southern Ethiopia in 2010 [40]. In the Ethiopian context, the average household out pocket expenditure from the total health financing is around 35% [41, 42]. Based on this rate, the starting price we expected to be paid was 1000 Birr (0.35x2824birr).

After presenting the case scenario, all participants were asked:

***Are you willing to pay some positive price for cataract surgery?***

*1. Yes                          2. No*

Those who said no or unwilling to pay for the service were dropped for the next bidding game, which was used to assess the participants' maximum willingness to pay for cataract surgery. But those who said yes for the first question were asked the maximum price they want to pay by the for the cataract surgery (**Fig 1**).

**Pre-operative visual ability/visual function.** The visual system's function can be measured objectively or subjectively. Visual acuity measurement is one of the objective indicators of the visual system functionality. However, it does not indicate the overall visual performance of an individual. Hence, we used the subjective measurement, the visual function/visual ability, to assess the overall visual performance. Hence it was measured by using adapted Visual Function 14 Index (VF-14) tool. Previously, numerous researchers used the tool as a standard. However, based on the socioeconomic and cultural perspective of their study population, the tool was adapted. The response option was prepared with 5-point Likert's scale format. The scale ranged from the 0–4 value for the degree of visual difficulties' unable to do' (scale 0), 'great difficulty' (scale1), 'moderate difficulty' (scale 2), 'little difficulty' (scale 3) and 'no difficulty' (scale 4). Then each scale was multiplied by 25. So, the value of each response ranges from 0–100. After that, the factored amounts were summed up. Finally, VF index was computed by dividing the summed factored amounts to the number of checked boxes and the visual ability were categorized [43–45].

The calculation is elaborated across the studies, and the final classification is based on the International Council of Ophthalmology as normal or near-normal performance ($\geq$50 visual ability score) and restricted performance ($<$50 visual ability score) [46].

**Wealth index.** Household wealth was measured by asking several questions on the possession of different agricultural products, non-productive assets, and household facilities. The wealth index quintile was derived by Principal Component Analysis (PCA) [47, 48].

**Data processing and analysis.** The data were coded and entered into Epi Info version 7 and was exported to Stata Version 14 for analysis. The Tobit econometric model was used to

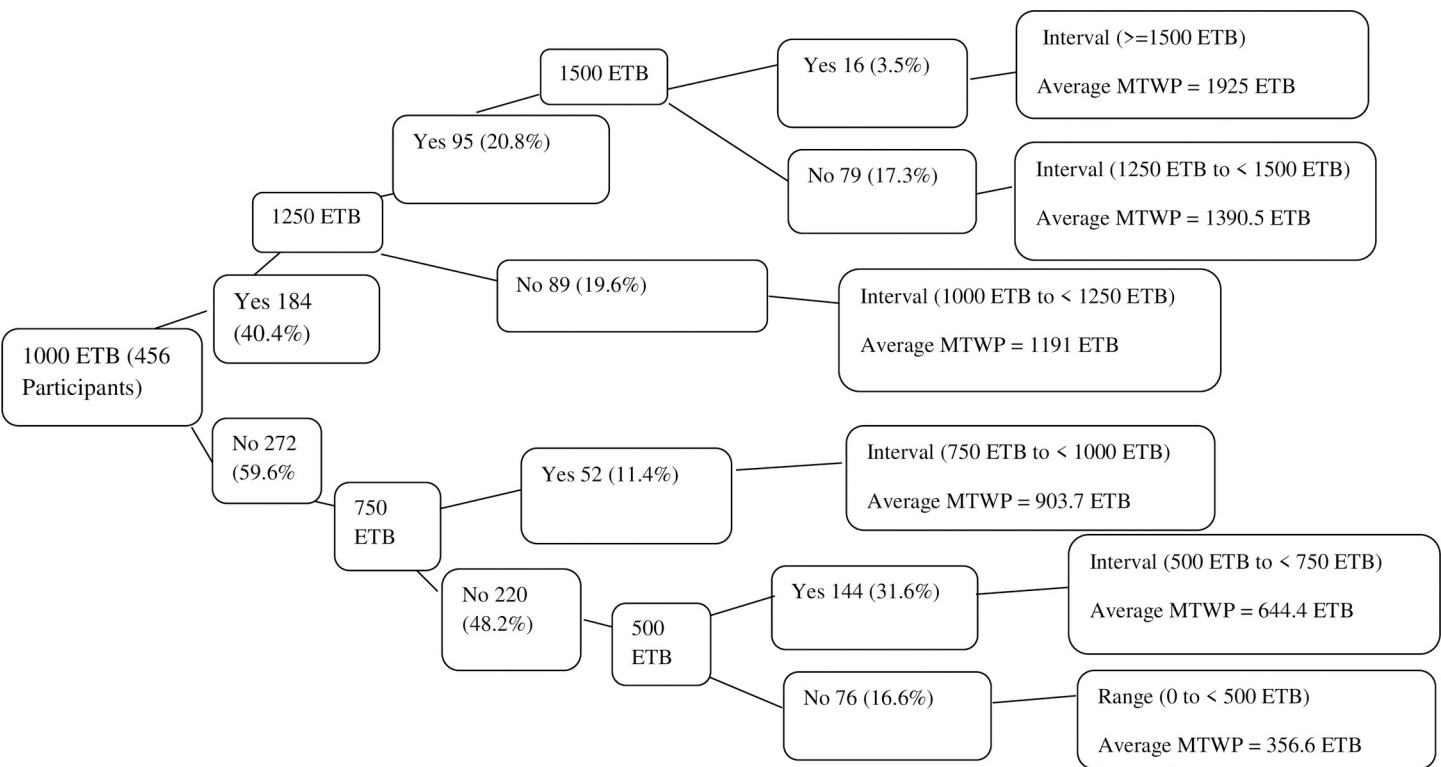

**Fig 1. Iterative bidding technique to elicit willingness to pay for cataract surgery in outreach Sites, North West Ethiopia.** First bidding questions: If the cost of cataract surgery for one eye is ETB 1000, are you willing to pay that amount? December 2018 Exchange rate: US$ 1 = ETB 27.8.

analyze the determinants of willingness to pay and the maximum amount of money that patients willing to pay. This model is used to estimate parameters when the outcome variable is continuous in one end and has a constrained range in another arm [49–51]. The model is depicted as follow;

$$y = \begin{cases} 1 \\ 0 \end{cases}$$

$$i\ MWTP = Bo + B'\ Xi + e > 0$$

$$if\ MWTP \leq 0$$

Y = Outcome variable; MWTP = Maximum WTP; Xi = Explanatory variables; βo = Slope; β' = Coefficient; ε = error term; 1 = Success/Yes; 0 = Failure/No

The model estimates the marginal effect of an explanatory variable on the expected value of the dependent variable. The descriptive data were organized and presented using summary statistics, frequency distribution tables, and figures accordingly. The Tobit model assumptions [52, 53] like normality, Multicollinearity (VIF = 1.23), and homoscedasticity (Breusch-pagan (hettest, $X^2 = 1.78$, p = 0.18)) of error terms were checked. Regression coefficients (β), 95% confidence interval, and p-value of $< 0.05$ were used to measure a statistical association's strength and presence.

### Ethical considerations

Ethical clearance was obtained from the ethical committee of the Institute of Public Health, College of Medicine and Health Science, and the University of Gondar. An official letter of permission was also obtained from Amhara Regional Health Bureau and Bahir Dar Tibebe Gion Specialized Referral Hospital. After explaining the purpose and importance of the study, informed consent was obtained from each study participant. Participants were also informed that they have the right to withdraw from the study if they face any inconvenience during the interview processes.

## Results

### Socio-demographic and economic characteristics of participants

A total of 827 participants responded to the interviewer-administered questionnaire with a response rate of 89%. The median age of the participants was 65 years, with an interquartile range of 15. About 55% of the participants were age $\geq$ 65 years. The majority of the participants, 498 (60.2%), were males. The significant portion of the participants, 675(82%), were from rural and had no formal education of 786(95%). Most of the participants were currently married 515(62.2%). About 84% of the participants had a family head role. About two-thirds of the participants, 565 (68.3%), were in a state of still working (**Table 1**).

### Health-related characteristics of participants

The majority of the participants, 638 (77.1%), were not familiarized with cataract surgery either from themselves or their family members. Nearly 714(86%) of the participants had unilateral operable cataracts. Preoperatively, the participants' mean visual ability score was 42.38, with a standard deviation of 23. The majority of the participants, 536(65%), had restricted visual performance, that was less than 50 visual ability score. The mean time spent with the condition was 2years. About 787(95%) of the participants did not report ocular co-morbidity other than cataract. Likewise, most of the participants, 720(87%), did not report any systemic illness. More than half of the participants, 466(56.3%), stated that they utilized community-based health insurance services (**Table 2**).

### Participants' willingness to pay for cataract surgery

Fifty-five percent of the participants(456) were willing to pay something for the cataract surgery, and the average amount of money participants willing to pay was 500 ETB or 17.9 USD (10.9, 28.77).

It was calculated as follows:-

$$\text{AMWTP} = \frac{\sum 827(\text{MWTP1} + \text{MWTP2} + \cdots + \text{MWTP827})}{827}$$
$$= 500 \ birr \text{ for cataract surgery service}$$

*Note—1$USD = 27.9 birrs in May 2019 currency exchange, AMWTP is an average maximum willingness to pay*

### Associated factors for willingness to pay some positive price for cataract surgery

The study revealed that the eye involved with cataract, marital status, and occupation as significant variables for participants' willingness to pay some positive price for cataract surgery.

**Table 1. Socio-demographic characteristics of the study participants for WTP for cataract surgery in outreach Site, Northwest Ethiopia, 2019 (n = 827).**

| Variables | Willing to Pay | | | | Total (827) | |
|---|---|---|---|---|---|---|
| | Yes (456) | | No (371) | | | |
| | Frequency | % | Frequency | % | Frequency | % |
| **Sex** | | | | | | |
| Male | 292 | 35.3 | 206 | 24.9 | 498 | 60.2 |
| Female | 164 | 19.8 | 165 | 20 | 329 | 39.8 |
| **Age** | | | | | | |
| <65 years | 220 | 26.6 | 154 | 18.6 | 374 | 45.2 |
| ≥65years | 236 | 28.5 | 217 | 26.3 | 453 | 54.8 |
| **Residence** | | | | | | |
| Rural | 373 | 45.1 | 302 | 36.5 | 675 | 81.6 |
| Urban | 83 | 10.1 | 69 | 8.3 | 152 | 18.4 |
| **Marital status** | | | | | | |
| Married | 315 | 38.1 | 200 | 24.2 | 515 | 62.3 |
| Widowed | 111 | 13.4 | 118 | 14.3 | 229 | 27.7 |
| Divorced | 21 | 2.5 | 42 | 5.1 | 63 | 7.6 |
| Single | 9 | 1.1 | 11 | 1.3 | 20 | 2.4 |
| **Educational status** | | | | | | |
| Noformal education | 430 | 52 | 356 | 43 | 786 | 95 |
| Formal education | 26 | 3.1 | 15 | 1.9 | 41 | 5 |
| **Working state** | | | | | | |
| Still working | 349 | 42.2 | 216 | 26.1 | 565 | 68.3 |
| Retired/unemployed | 107 | 13 | 155 | 18.7 | 262 | 31.68 |
| **Household head** | | | | | | |
| Yes | 387 | 46.8 | 308 | 37.2 | 695 | 84 |
| No | 69 | 8.4 | 63 | 7.6 | 132 | 16 |
| **Family size** | | | | | | |
| <5 | 305 | 36.9 | 281 | 34 | 586 | 70.86 |
| ≥5 | 151 | 18.3 | 90 | 10.8 | 241 | 29.14 |
| **Wealth status** | | | | | | |
| Poor | 155 | 18.7 | 120 | 14.5 | 275 | 33.3 |
| Medium | 139 | 16.8 | 137 | 16.6 | 276 | 33.4 |
| Rich | 162 | 19.6 | 114 | 13.8 | 276 | 33.4 |

Study participants whose one eye involved with cataracts were 36% (AOR = 0.64; 95% CI: 0.41, 0.98) less likely to pay some positive price for cataract surgery compared to those whose both eyes were involved. Besides, those who are not currently married were 1.55 (AOR = 2.55; 95% CI: 1.12, 2.14) times more likely to pay some positive cataract surgery price. The odds of participants who were still in working condition were 51% (AOR = 0.49; 95% CI: 0.35, 0.69) less likely to pay some positive price for cataract surgery than those who were not working (**Table 3**).

## Associated factors for willingness to pay for cataract surgery and their marginal effects

The study showed that the still working participants were willing to pay 26.66USD more than the participants who retired or unemployed, keeping other variables constant (β = 26.66,95% CI13.03, 40.29). This variable's marginal effect revealed that changing the working state from

Table 2. Health-related characteristics of the study participants for WTP for cataract surgery in outreach Site, North West Ethiopia, 2019 (n = 827).

| Variables | Willing to pay | | | | Total | |
| --- | --- | --- | --- | --- | --- | --- |
| | Yes | | No | | | |
| | Frequency | % | Frequency | % | Frequency | % |
| **Familiarity with cataract surgery** | | | | | | |
| Yes | 106 | 12.8 | 83 | 10.1 | 189 | 22.9 |
| No | 350 | 42.3 | 288 | 34.8 | 638 | 77.1 |
| **Eye involved with cataract** | | | | | | |
| One eye | 408 | 49.3 | 306 | 37 | 714 | 86.3 |
| Both eyes | 48 | 5.8 | 65 | 7.9 | 113 | 13.7 |
| **Preoperative Visual ability** | | | | | | |
| <50 | 291 | 35.2 | 245 | 29.6 | 536 | 64.8 |
| ≥50 | 165 | 20 | 126 | 15.2 | 291 | 35.2 |
| **Time spent with condition** | | | | | | |
| <2years | 147 | 17.8 | 96 | 11.6 | 243 | 29.4 |
| ≥2years | 309 | 37.4 | 275 | 33.2 | 584 | 70.6 |
| **Self-reported ocular morbidity** | | | | | | |
| Yes | 16 | 1.9 | 24 | 2.9 | 40 | 4.8 |
| No | 440 | 53.2 | 347 | 42 | 787 | 95.2 |
| **Self-reported Systemic illness** | | | | | | |
| Yes | 55 | 6.7 | 52 | 6.3 | 107 | 12.9 |
| No | 401 | 48.4 | 319 | 38.6 | 720 | 87.1 |
| **Health insurance** | | | | | | |
| Yes | 276 | 33.4 | 190 | 22.9 | 466 | 56.3 |
| No | 180 | 21.8 | 181 | 21.9 | 361 | 43.7 |

retired/unemployed to active working state increases WTP for cataract surgery by 14.46 USD from mean willingness to pay (dy/dx = 14.46, 95% CI 7.48, 21.51). Participants who had formal education were WTP 29.16 USD more than participants who had no formal education (β = 29, 16, 95%CI 2.35, 55.97).

Participants who had no ocular-comorbidity were willing to pay 28.48USD more than those with ocular co-morbidity, holding other variables constant (β = 28.48, 95%CI1.08, 55.90). The study depicted that being free from ocular morbidity increases WTP for cataract surgery by 14.26 USD from mean willingness to pay (dy/dx = 14.26, 95%CI 2.48, 26.04).

Likewise, participants with unilateral cataracts were WTP 21.21USD more than those with bilateral cataracts, holding other variables constant (β = 21.21, 95%CI3.65, 38.77). This variable's marginal effect showed that only having unilateral cataract increases WTP for cataract surgery by 11.20 USD from mean willingness to pay (dy/dx = 11.20, 95% CI 2.68, 19.73).

It was also depicted that a one-year increment in years spent with the condition, the participants' WTP decreases by 1.69 USD, keeping the other variables constant (β = -1.69,95%CI -3.32, -0.07). This variable's marginal effect indicated that increasing years lived with the condition by one year, it may decrease the WTP for cataract surgery by 1.12 USD from the mean (dy/dx = -1.12, 95%CI -2.05, -0.19).

As the better eye's visual ability increases by one unit, the participants' WTP decreases by 0.29USD (β = -0.29, 95%CI -0.55, -0.04). This variable's marginal effect revealed that increasing the visual ability in the better by one unit can decrease the WTP for cataract surgery by 0.16USD from mean willingness to pay (dy/dx = -0.16,95% CI-0.31, -0.02) **(Table 4)**.

**Table 3. Logistic regression for associated factors for willingness pay some positive price for cataract surgery in outreach Site, North West Ethiopia, 2019 (n = 827).**

| Variable | Category | Willing to pay | | COR (95% CI) | AOR (95% CI) | P-value |
|---|---|---|---|---|---|---|
| | | Yes | No | | | |
| Age | < 65 years | 220 | 154 | 0.76 (0.58, 1.00) | 0.86 (0.63, 1.19) | 0.37 |
| | > = 65 years | 236 | 217 | 1 | | |
| Family head | Yes | 387 | 308 | 0.87 (0.60, 1.67) | 0.70 (0.47, 1.06) | 0.09 |
| | No | 69 | 63 | 1 | | |
| Residence | Rural | 373 | 302 | 1 | | |
| | Urban | 83 | 69 | 1.03 (0.72, 1.46) | 0.90 (0.59, 1.37) | 0.63 |
| Familiarity with cataract surgery | Yes | 106 | 83 | 1 | | |
| | No | 350 | 288 | 1.05 (0.76, 1.46) | 1.29 (0.71, 1.42) | |
| Self-reported ocular co-morbidity | Yes | 16 | 24 | 1 | | |
| | No | 440 | 347 | 0.53 (0.28, 1.01) | 0.54 (0.27, 1.06) | 0.07 |
| Eye involved with cataract | One | 408 | 306 | **0.55 (0.37,0.83)** | **0.64 (0.41, 0.98)** | **0.04**\* |
| | Both | 48 | 65 | 1 | | |
| Self-reported systemic co-morbidity | Yes | 55 | 52 | 1 | | |
| | No | 401 | 319 | 0.84 9)0.56. 1.26 | 0.95 (0.61, 1.47) | 0.82 |
| Health Insurance | Yes | 276 | 190 | 0.68 (0.52, 0.90) | 0.82 (0.61, 1.11) | 0.20 |
| | No | 180 | 181 | 1 | | |
| Preoperative Visual Ability | < 50 | 291 | 245 | 1 | | |
| | > = 50 | 165 | 126 | 0.91 (0.68, 1.21) | 0.80 (0.59, 1.10) | 0.17 |
| Time spent with cataract | < 2 years | 147 | 96 | 0.73 (0.54, 0.99) | 1.24 (0.90, 1.71) | 0.19 |
| | > = 2 years | 309 | 275 | 1 | | |
| Wealth status | Poor | 155 | 120 | 1 | | |
| | Medium | 139 | 137 | 1.27 (0.91, 1.78) | 1.31 (0.91, 1.91) | 0.15 |
| | Rich | 162 | 114 | 0.91 (0.65, 1.27) | 1.03 (0.72, 1.47) | 0.87 |
| Marital status | Married | 315 | 200 | 1 | | |
| | Not-married | 141 | 171 | **1.91 (1.44, 2.54)** | **1.55 (1.12, 2.14)** | **0.008**\* |
| Educational status | No formal education | 430 | 356 | 1 | | |
| | Formal education | 26 | 15 | 0.70 (0.36, 1.34) | 0.65 (0.32, 1.33) | 0.24 |
| Occupational status | Retired/unemployed) | 349 | 216 | 1 | | |
| | Still working | 107 | 155 | **0.43 (0.32, 0.58)** | **0.49 (0.35, 0.69)** | **0.001**\* |
| Family size | < 5 | 305 | 281 | 1.55 (1.34, 2.10) | 1.29 (0.92, 1.80) | 0.14 |
| | > = 5 | 151 | 90 | 1 | | |
| Constant | | | | | 3.89 | |

## Tobit/OLS/Truncated econometric analysis of factors associated with WTP for cataract surgery

Tobit, OLS, and Truncated analysis were performed for factors related to WTP for cataract surgery. Educational status, working condition, visual ability, time spent with the disease, eye involved with cataract, and self-reported ocular co-morbidity were significant variables for WTP for cataract surgery by the Tobit model. In the OLS model, educational status, visual ability, and cataract eyes were significantly associated with WTP for cataract surgery. Besides, participants' educational level and visual ability were the significant variables for WTP (**Table 5**).

## Discussion

This study aimed to assess WTP for cataract surgery and associated factors to provide substantive evidence for developing a cost-recovery model that assures self-sustaining and high-

**Table 4. Maximum likelihood of Tobit econometric analysis of factors associated with WTP for cataract surgery in outreach Site, North West Ethiopia,2019 (n = 827).**

| Variables for MWTP | Category | β coefficients | SE | t-value | p-value | 95% CI Lower, upper | Dy/dx(95% CI) |
|---|---|---|---|---|---|---|---|
| Age | N | -0.34 | 0.26 | -1.30 | 0.19 | -0.85, 0.17 | 0.19(-0.48, .098) |
| Being Married (ref, single) | D | 9.43 | 6.83 | 1.38 | 0.17 | -3.98, 22.84 | 5.31(-2.14, 12.78) |
| Being Urban (ref, rural) | D | -0.77 | 8.24 | -0.09 | 0.93 | -16.95, 15.42 | 2.94 (-5.80, 11.68) |
| Formal education (ref, no formal education) | D | 29.16 | 13.66 | 2.14 | **0.03**\* | 2.35, 55.97 | 18.62(-0.22,37.46) |
| Still working (ref, retired/unemployed) | D | 26.66 | 6.94 | 3.84 | **0.001**\* | 13.03, 40.29 | **14.49 (7.48, 21.51)** \* |
| Being Family head(ref, No) | D | 5.25 | 8.10 | 0.65 | 0.517 | -10.65, 21.15 | -0.44 (-9.61, 8.74) |
| Visual ability | N | -0.29 | 0.13 | -2.26 | **0.024**\* | -0.55, -0.04 | **-0.16(0.31, -0.02)** \* |
| Time spent with the condition | N | -1.96 | 0.83 | -2.37 | **0.018**\* | -3.59, 0.34 | **-1.12 (-2.05, -0.19)** \* |
| Familiarity with cataract surgery (ref, yes) | D | 0.68 | 6.79 | 0.10 | 0.920 | -12.67, 14.03 | 0.39 (-7.18, 7.96) |
| No ocular co-morbidity (ref, yes) | D | 28.49 | 13.96 | 2.04 | **0.04**\* | 1.08, 55.90 | **14.26 (2.48, 26.04)** \* |
| No systemic illness (ref, yes) | D | 0.77 | 8.65 | 0.09 | 0.93 | -16.2, 17.76 | 0.43 (-9.18, 10.06) |
| Involved eye with cataract(ref, both eyes) | D | 21.21 | 8.95 | 2.37 | **0.018**\* | 3.65, 38.77 | **11.20 (2.68,19.73)** \* |
| Medium ref, poor) | D | -12.44 | 7.3 | -1.70 | 0.089 | -26.76, 1.89 | -7.05(-15.16, 1.05) |
| Rich (ref, poor) | D | -4.51 | 6.90 | -0.65 | 0.51 | -18.07, 9.04 | -2.65 (-10.62, 5.31) |
| Having Health insurance(ref, No) | D | 9.52 | 6.02 | 1.58 | 0.114 | -2.28, 21.33 | 5.39 (-1.25, 12.04) |
| Family size | N | 1.52 | 1.73 | 0.88 | 0.38 | -1.87, 4.91 | 0.86 (-1.06, 2.79) |
| **Constant** | | -29.02 | 28.07 | -1.03 | 0.30 | -84.12, 26.07 | |
| **/Sigma** | | 73.08 | 2.68 | | | 67.82 78.34 | |

D, Dummy variable; N, numeric variable; ref, reference category;

\* significant with p-value <0.05.

volume surgical services. Although more than half of the patients admitted for cataract surgery (55%) were willing to pay something for the surgery, the average amount of money ready to pay for the surgery per eye was 500ETB (17.9USD).

**Table 5. Tobit/OLS/Truncated econometric analysis of factors associated with WTP for cataract surgery in outreach Site, North West Ethiopia, 2019 (n = 827).**

| Variables | Category | Tobit Model | | | OLS Model | | | Truncated Model | | |
|---|---|---|---|---|---|---|---|---|---|---|
| | | β coefficients | SE | P-value | β coefficients | SE | P-value | β coefficients | SE | P-value |
| Age | N | -0.34 | 0.26 | 0.19 | -0.23 | 0.15 | 0.13 | -0.45 | 0.30 | 0.13 |
| Being Married (ref, single) | D | 9.43 | 6.83 | 0.17 | 2.57 | 4.01 | 0.56 | -11.9 | 7.72 | 0.12 |
| Being Urban (ref, rural) | D | -0.77 | 8.24 | 0.93 | -1.8 | 4.86 | 0.71 | -9.1 | 9.17 | 0.32 |
| Formal education (ref, no formal education) | D | 29.16 | 13.66 | **0.03**\* | 22.7 | 8.28 | **0.006**\* | 30.3 | 13.49 | **0.03**\* |
| Still working (ref, retired/unemployed) | D | 26.66 | 6.94 | **0.001**\* | 12.5 | 4.02 | 0.002 | -0.46 | 8.20 | 0.96 |
| Being Family head(ref, No) | D | 5.25 | 8.10 | 0.517 | 0.36 | 4.78 | 0.94 | -11.02 | 8.92 | 0.22 |
| Visual ability | N | -0.29 | 0.13 | **0.024**\* | -0.79 | 0.08 | **0.008**\* | -0.35 | 0.14 | **0.01**\* |
| Time spent with the condition | N | -1.96 | 0.83 | **0.018**\* | -0.79 | 0.46 | 0.09 | 1.1 | 0.96 | 0.29 |
| Familiarity with cataract surgery (ref, yes) | D | 0.68 | 6.79 | 0.920 | -0.46 | 4.05 | 0.91 | -1.8 | 7.53 | 0.82 |
| No ocular co-morbidity (ref, yes) | D | 28.49 | 13.96 | **0.04**\* | 13.4 | 7.81 | 0.09 | 11.4 | 17.53 | 0.52 |
| No systemic illness (ref, yes) | D | 0.77 | 8.65 | 0.93 | 0.25 | 5.10 | 0.96 | -2.3 | 9.88 | 0.81 |
| Involved eye with cataract(ref, both eyes) | D | 21.21 | 8.95 | **0.018**\* | 11.9 | 5.14 | **0.02**\* | 12.7 | 10.86 | 0.24 |
| Medium ref, poor) | D | -12.44 | 7.3 | 0.089 | -6.9 | 4.33 | 0.11 | -0.46 | 8.09 | 0.52 |
| Rich (ref, poor) | D | -4.51 | 6.90 | 0.51 | -3.6 | 4.15 | 0.38 | -6.1 | 7.60 | 0.42 |
| Having Health insurance(ref, No) | D | 9.52 | 6.02 | 0.114 | 6.7 | 3.55 | 0.06 | 8.2 | 6.79 | 0.23 |
| Family size | N | 1.52 | 1.73 | 0.38 | 0.92 | 1.03 | 0.38 | 0.10 | 1.89 | 0.96 |
| **Constant** | | -29.02 | | | 28.4 | | | 94.2 | 31.3 | |

This result was comparable with outreach based studies done in Jimma, Ethiopia (US$12.4 [22], and in Northern Nigeria (US$18.5) [19]. On the other hand, it is greater than the campaign based studies conducted in Tanzania (US$2.3 [20], Malawi (3US$), [21], and Nepal (US $7) [17]. Despite design similarity, they were conducted on a minimal sample size selected with non-probability sampling. This may be a reason for the discrepancy. However, this result was lower than the studies done in Hong Kong (US$ 552) [23] and China (USD 968) [25]. Both studies involved cataract patients on the waiting list in hospitals and used a bidding format for eliciting the willingness to pay. Relatively, their target price was higher as compared to the present study. Having a difference in target price and research setting might be taken as a reason for the variation.

Overall, this amount of money willing to pay matches the subsidized price for cataract surgery at General and Tertiary Hospitals. However, it was significantly lower than the current cost of the surgery area, 2824 ETB(101.56USD) [40], and the proposed target price by 50%. Lack of money, lack of knowledge to access the services, and waiting for the campaign were the main reason cited to be unwilling to pay and unable to seek eye care services previously.

Currently, cataract surgery is being delivered for districts through outreach programs, with the tremendous aid of donors, free of charge. Nevertheless, this might not be financially sustainable for the long-run. This study's findings imply that if the government or private sectors plan to sustain the program with domestic resources, it is strategic to develop an inclusive cost-recovery model that can assure services accessibility, self-reliance, and equity. The Aravind Eye Care System in India is the best lesson: providing high-quality and high-volume (350,000 eye operations per year) services to those who can afford to pay market rates and then uses the profits to fund care for those who cannot. Patients who cannot afford to pay are given cataract surgery for free. However, the government reimburses Aravind US$10 for each procedure [54]. The Ethiopian health care financing policy considers the systematizing fee waiver system, while such approaches are not applied for cataract surgical services so far.

Moreover, the WTP's comparability with the current price of cataract surgery set for General and Tertiary Hospitals indicates that there is a probability of integrating the service in the existing health care system at the district level if the Ethiopian government invests on human resource development and infrastructure.

The working state's impact on willingness to pay is depicted that the participants who were still working were WTP 26.66USD more than the participants who were retired/unemployed. This result is supported by a study conducted in rural china [15] though some studies concluded that willingness to pay was not affected by the working state [18, 21, 25]. This relationship can be explained in a way that employees can be related to greater productivity. This implies that retired/ unemployed persons would be a potential target for cost- recovery.

Persons with formal education were 29.16 USD more WTP than those who had no formal education for surgery. This finding is consistent with a study done in China [24]. This may suggest that educated persons may value their vision more and may have good knowledge about the service's cost-benefit aspects. However, many studies found that WTP for cataract surgery was not significantly affected by literacy because these studies were done in nations where there is higher educational coverage [15, 18, 22, 23]. Hence, significant variation based on education might not be observed. This result indicated that extra efforts in outreach programs need to be invested in prospering illiterate persons' knowledge with non-printed media.

Patients with no ocular-comorbidity and unilateral cataracts were more willing to pay than those with ocular co-morbidity and bilateral cataract. Previous studies also reported that pre-existing eye diseases and bilateral cataracts affect willingness to pay for cataract surgery [25]. This may be explained so that persons with co-morbidity and bilateral cataract might have poor visual performance. Hence, their productivity and economy may be compromised,

making them desperate to have good vision and uptake the services. On the other hand, as the visual performance of a better eye decreases by one unit, the participants' WTP increases by 0.29USD.

Similarly, a study reported that persons with poor visual performance were more willing to pay for the surgery [23]. The justification might be that patients with poor visual ability can be highly ambitioned to enhance their daily base activities regardless of their ability to pay. On the contrary, two studies found that as the preoperative visual acuity becomes more impaired, the willingness to pay decreases. The visual functionality was measured objectively [21, 24]. These findings imply that cataract surgery's subsidization is substantive for patients with a severe visual disability, ocular-comorbidity preexisted condition, and bilateral cataract. Thus, even if resource utilization efficiency is sacrificed, equity among services consumers might be logical.

The study also depicted that one year increment in years lived with visual impairment, the participants WTP decreases by 1.69 USD. The explanation might be that as the duration increase without intervention, the severity of visual disability and complication associated with cataract can be increased. This increases the psycho-socio-economic burden of the condition. Hence, it ends up with a poor vision-related quality of life and poor ability to pay. This implies that early intervention may promote a greater willingness to pay and service utilization.

## Limitation of the study

The method used only shows the service's values or benefits, which didn't consider the full economic evaluation approach. The other limitation of this study is the strategic response bias in which the respondents will intentionally overestimate or underestimate the price, which is not their actual price. We tried to minimize this bias using a well-stated case scenario and Bidding game format as an elicitation approach.

## Conclusion

Willingness to pay for cataract surgery among adults with operable cataracts in outreach sites is much lower than the surgery's actual cost. People with a state of having a formal education, working state, less time spent with the condition, free from co-morbidity, and bilateral cataracts were more willing to pay for cataract surgery. Therefore, the provision of cataract surgical services by offering a multi-tiered package attributed to the population's characteristics is essential and applicable.

## Supporting information

**S1 File.**
(DOCX)

**S1 Questionnaire.**
(DOCX)

**S1 Data.**
(DTA)

## Acknowledgments

The authors would like to thank all respondents for their willingness to participate in the study. We are also grateful to the Amhara public health institute, Tebebe Gion Specialized

Referral Hospital, and the University of Gondar for material support. Finally, our appreciation goes to data collectors for their unreserved contribution in data collection activities.

## Author Contributions

**Conceptualization:** Mohammed Seid.

**Data curation:** Mohammed Seid.

**Formal analysis:** Mohammed Seid, Amare Minyihun, Gizachew Tilahun, Asmamaw Atnafu, Getasew Amare.

**Methodology:** Mohammed Seid, Amare Minyihun, Gizachew Tilahun, Asmamaw Atnafu, Getasew Amare.

**Software:** Mohammed Seid, Amare Minyihun, Gizachew Tilahun, Asmamaw Atnafu, Getasew Amare.

**Writing – original draft:** Mohammed Seid.

**Writing – review & editing:** Amare Minyihun, Asmamaw Atnafu, Getasew Amare.

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
