## [Decision Letter · Decision Letter 0]

13 May 2020

PONE-D-19-31270

Willingness to Pay for Cataract Surgery and Associated Factors among Cataract Patients in Outreach Site, North West Ethiopia

PLOS ONE

Dear Mr. Amare,

Thank you for submitting your manuscript to PLOS ONE. After careful consideration, we feel that it has merit but does not fully meet PLOS ONE’s publication criteria as it currently stands. Therefore, we invite you to submit a revised version of the manuscript that addresses the points raised during the review process.

We would appreciate receiving your revised manuscript by Jun 27 2020 11:59PM. To enhance the reproducibility of your results, we recommend that if applicable you deposit your laboratory protocols in protocols.io, where a protocol can be assigned its own identifier (DOI) such that it can be cited independently in the future. For instructions see: http://journals.plos.org/plosone/s/submission-guidelines#loc-laboratory-protocols

We look forward to receiving your revised manuscript.

Kind regards,

Janhavi Ajit Vaingankar

Academic Editor

PLOS ONE

Journal Requirements:

2. Please address the following:

- Please include additional information regarding the survey or questionnaire used in the study and ensure that you have provided sufficient details that others could replicate the analyses. For instance, if you developed a questionnaire as part of this study and it is not under a copyright more restrictive than CC-BY, please include a copy, in both the original language and English, as Supporting Information.

- Please ensure you have thoroughly discussed any potential limitations of this study within the Discussion section.

- Please refrain from reporting p values as 0.000, either report the exact value or use the format p<0.0001.

Thank you for your attention to these queries.

Additional Editor Comments (if provided):

Reviewers' comments:

Reviewer's Responses to Questions

**Comments to the Author**

1. Is the manuscript technically sound, and do the data support the conclusions?

Reviewer #1: Yes

Reviewer #2: Partly

Reviewer #3: Partly

2. Has the statistical analysis been performed appropriately and rigorously? 

Reviewer #1: N/A

Reviewer #2: I Don't Know

Reviewer #3: I Don't Know

3. Have the authors made all data underlying the findings in their manuscript fully available?

Reviewer #1: Yes

Reviewer #2: Yes

Reviewer #3: No

4. Is the manuscript presented in an intelligible fashion and written in standard English?

Reviewer #1: No

Reviewer #2: No

Reviewer #3: No

5. Review Comments to the Author

Reviewer #1: Access to healthcare is a fundamental human right. Understanding the willingness to pay for cataract surgery and the associated factors are crucial in designing self-sustaining and efficient eyecare programs. I applaud the authors for contributing to the body of evidence.

However, as there was no line number given in the manuscript, sticky-notes were used (by the reviewer) to provide specific comments on the manuscript. It is highly recommended to use line numbers in the next versions. The sticky notes contain comments targeted to specific sentences or wordings; whereas the suggestions shared here are more general in nature. Authors are suggested to first go through the general-comments and then check the sticky-note-comments.

Suggestions for improvement are as follows:

1. The manuscript (including the abstract) demands massive improvement in writing. Moreover, there are typos and grammatical errors that need to be corrected.

2. WTP are health-state dependent, hence, it depends on the context and techniques applied (e.g. double bounded dichotomous choices) to elicit WTP. Sharing the tool (as supplementary documents) might be a good to option to let the reader understand the WTP estimation techniques applied in this study.

3. Generally outreach-sites are socio-demographically different from the general population. A comparison of socio-demographic characteristics between the sample and general population of North-West-Ethiopia or Ethiopia might provide a better understanding of the sample as-well-as understanding the differences in WTP in general context.

4. The ‘Introduction’ mentions about the mean costs of cataract surgery in Europe and USA. The cost of the surgery varies across the globe. As the study estimates the WTP for cataract surgery in Ethiopia, it would be wise to also state literatures that estimate WTP of other African and Asian countries (similar economies).

5. The ‘Introduction’ ends with a non-justifiable argument (Please check the sticky-note-comment). The manuscript states about the context-dependent WTP, however, it does not state any literature on context or health-state dependencies of estimated WTP. Also, please provide a detailed description of the context (i.e. WTP elicitation approach) in the method section.

6. There are various surgical techniques of cataract surgery (e.g. Small incision cataract surgery, phacoemulsification). For which technique of surgery respondents elicited their WTP for? Did they have full information or any prior knowledge on the surgical procedure(s)?

7. The ‘Study design and setting’ section should focus on the eyecare provided by the hospital. Please provide a summary of the eyecare facilities available in North West Ethiopia. It will help the reader to understand the representativeness of the setting (i.e. hospital) from which respondents were selected.

8. The ‘Study population and sampling technique’ section needs to improve its clarity in describing the sampling technique used for the study. Simple random sampling technique requires selecting subjects from a predetermined sampling frame. If the list of 1336 cataract patients was used as the sampling frame, then at least two things needs to be clarified:

(a) The sampling frame was prepared based on the patients visited the facility from June 10, 2018 to June 17, 2018; were the subjects chosen on a daily basis or based on an aggregated list.

(b) As the program provides free of cost cataract surgery, the subjects interviewed in this study should already have a preconceived estimation about the cost of the treatment; this has a definite impact on the elicited WTP (e.g. only 55 percent respondents elicited a non-zero WTP).

9. The description of bidding procedure given in page 5 needs clarity. How was initial bid (i.e. 1000 ETB) was chosen? A schematic representation of the DBDC experiment would help reader to visualize the context better. Please check the following reference:

[Islam M.N., Engles T., Hossain S., Sarker M., Rabbani A. “Willingness-to-pay for cataract surgeries among patients visiting eye-care facilities in Dhaka.” Applied Health Economics and Health Policy 17, no. 4 (2019): 545-554. DOI: 10.1007/s40258-019-00478-3]

10. About 45 percent respondents were not willing to pay a positive price for the service, how were these responses handled in the analysis? In addition, if some people were not willing to pay, were these people statistically different from those that were willing to pay?

11. The DBDC experiment ends with an interval of WTP. Did you check the interval regression techniques to estimate the associated factors? (If yes) Were the results different with an interval regression model?

12. The average WTP can be estimated using a constant only interval regression model. Although, 45 percent respondents did not elicit a positive WTP, their true WTP might be greater than zero but less than the last bid (750 ETB). In that case counting them as zero (as done in this manuscript) might not be pragmatic. Please check the following reference:

[Islam, M.N., Rabbani, A. & Sarker, M. “Health shock and preference instability: assessing health-state dependency of willingness-to-pay for corrective eyeglasses.” Health Economics Review 9, no. 1 (2019): 32. DOI:10.1186/s13561-019-0249-3]

13. The robustness of the Tobit models can be checked estimating coefficients of both censored and truncated samples as-well-as estimating the OLS coefficients. Please report the results (as supplementary documents). This practice will help the readers understand the robustness of the results obtained by the study.

14. The ‘Participant’s health-related characteristics’ section requires to provide sufficient details of the variables (what and how the measurements were taken; e.g. Systemic illness).

15. The discussion and conclusion need to be improved.

Reviewer #2: Dear Editor,

Thank you for inviting me to review this interesting article on willingness to pay for cataract surgery. The authors have attempted to tackle a very pertinent issue in appropriate pricing of cataract surgery. However, there are some concerns.

The overall standard of academic English is low and one has to struggle to read between the lines to understand the otherwise important information being conveyed. May I suggest the authors seek professional assistance is preparing the grammatical content of the manuscript.

Many of the references are out of date. There are more recent references on the prevalence and causes of blindness, globally.

In the Methods section, the authors state that all patients admitted for cataract surgery at Tebbe Gion Hospital formed the study population, then mentions that the patients were screened for cataract. What was the purpose of the screening?

When was this study conducted?

Several unfamiliar analytical terms are mentioned.

Double Bounded Dichotomous Choice Variant of the Contingent Valuation (CV)

Principal Component Analysis (PCA)

Homoscedasticity of error terms through Breush-pagan for Heterosekedasticity test

The authors should briefly explain what they are and put a reference as this is a broad remit medical journal.

The authors mention visual ability score; an unusual term to assess visual acuity. Standard visual acuity measures are available e.g. Snellens notation or the logMAR.

In summary, I believe the authors have an important message to convey, but this is obfuscated in poor grammar and complex analytical terminology.

Reviewer #3: I found it quite difficult to understand the text because the English is not written in standard syntax and grammar. I strongly suggest that the authors engage the service of a professional English editing service prior to re-review of the manuscript. In addition, the methods and statistics require some more explanation for clarity and ease of understanding by the reader.

6. PLOS authors have the option to publish the peer review history of their article (what does this mean?). If published, this will include your full peer review and any attached files.

Reviewer #1: Yes: Muhammed Nazmul Islam

Reviewer #2: Yes: Dr Ada E Aghaji

Reviewer #3: No

---

## [Author Response · Author response to Decision Letter 0]

22 Jul 2020

Response reviewer

Reviewer 01

Question 1. 

The manuscript (including the abstract) demands massive improvement in writing. Moreover, there are typos and grammatical errors that need to be corrected.

Authors’ response

Dear reviewer thanks for your comments and questions.

We tried to look and revise the whole document based on the given comments.

Question 02

WTP are health-state dependent, hence, it depends on the context and techniques applied (e.g. double bounded dichotomous choices) to elicit WTP. Sharing the tool (as supplementary documents) might be a good to option to let the reader understand the WTP estimation techniques applied in this study.

Authors’ response

Dear reviewer thanks for your comments and questions. 

We tried to include the WTP elicit schematic diagram on page 9.

Question 03

Generally, outreach-sites are socio-demographically different from the general population. A comparison of socio-demographic characteristics between the sample and general population of North-West-Ethiopia or Ethiopia might provide a better understanding of the sample as-well-as understanding the differences in WTP in a general context.

Authors’ response

Dear reviewer thanks for your comments and questions.

It is rewritten in a way that describes socio-demographic characteristics of the study population and for whom the study would be inferred. Stated in Page 6 as “Campaign Based cross-sectional study was conducted and the site was at Tebebe Gion Specialized Referral Hospital, Bahir Dar city, North West Ethiopia. There are only four Tertiary and general hospitals for about 14 million people in the catchment area, which provide comprehensive eye care services including cataract surgical services in North West Ethiopia. However, all of them are urban-centered and not accessible for more than 80% of the community. Hence, the outreach program was planned for communities where cataract surgical services are not available and for peoples in the disadvantaged area with low socio-economic status in North West Ethiopia. Initially, screening for cataracts was conducted in the nearby rural weredas before the campaign. During the screening, Cataract cases were diagnosed and informed about the condition and scheduled to attend the planned outreach program. Hence, the cases were selected and linked to the campaign from weredas of West Gojjam, South Gondar, and Awi Zone. A total of 1336 cases (628 cases from West Gojjam, 374 cases from South Gondar, 201cases from Awuyi zone, 100 cases from Bahir Dar City, and 33 cases from East Gojjam) were admitted to the outreach Site. Thus more than 93% of cases were from rural weredas and district towns. Therefore, the findings of this study could apply to rural residents admitted for cataract surgery in outreach programs”

Question 04

The ‘Introduction’ mentions the mean costs of cataract surgery in Europe and USA. The cost of the surgery varies across the globe. As the study estimates the WTP for cataract surgery in Ethiopia, it would be wise to also state literatures that estimate WTP of other African and Asian countries (similar economies).

Authors’ response

Dear reviewer thanks for the comments and questions.

It is rewritten in pages 4-5 as “Moreover, studies from Africa and South East Asia reported that the actual cost exceeds the community’s willingness to pay for cataract surgery (15-20). Overall, the proportion of WTP in Asia was very discrepant, which was ranged from 34% to 90%. Similarly, the mean amount of money willing to pay was significantly varied from US$ 7 to US$ 968 (17, 19, 21, 22, 26). However, the mean amount of money WTP in Africa was ranged from US$2.3 to US$18.5, which was low as compared to Asian nations (15, 16, 20, 24)

Question 05

The ‘Introduction’ ends with a non-justifiable argument (Please check the sticky-note-comment). The manuscript states about the context-dependent WTP, however, it does not state any literature on context or health-state dependencies of estimated WTP. Also, please provide a detailed description of the context (i.e. WTP elicitation approach) in the method section.

Authors’ response

Dear reviewer thanks for the comments and questions

It is stated on page 5 as “Even if the knowledge regarding willingness to pay and the actual cost of the surgery is essential, the existed knowledge is not plenty to go through the intervention. Firstly, the existed knowledge in the study area is mainly centered on the proportion of willingness to pay rather than the amount of money (23). Secondly, the factors for the amount of money WTP were not fully addressed with the appropriate analysis model. Thirdly, willingness to pay is highly setting-variant, which mainly varies with socioeconomic status, and study setting. Consequently, it is difficult to apply the research finding done in other corners of the globe for this study area. Therefore, this study aimed to assess the willingness to pay for cataract surgery and associated factors among cataract patients at the outreach site in Northwest Ethiopia’’.

Question 06

There are various surgical techniques of cataract surgery (e.g. Small incision cataract surgery, phacoemulsification). For which technique of surgery respondents elicited their WTP for? Did they have full information or any prior knowledge of the surgical procedure(s)? 

Authors’ response 

Dear reviewer thanks for your comments and questions. 

It is stated on page 8 as “First, the participants were asked his/her willingness to pay something for the Manual Small incision cataract surgery.” Regarding prior information, they have been informed during screening, which is stated on page 6, as ‘During the screening, Cataract cases were diagnosed and informed about the condition…...”

Question 07

The ‘Study design and setting’ section should focus on the eye care provided by the hospital. Please provide a summary of the eye care facilities available in North West Ethiopia. It will help the reader to understand the representativeness of the setting (i.e. hospital) from which respondents were selected.

Authors’ response 

Dear reviewer thanks for your comments and questions.

It is written on page 6 as an “Outreach- based cross-sectional study was conducted. The campaign site was at Tebebe Gion Specialized Referral Hospital, Bahir Dar city, North West Ethiopia. There are only four Tertiary and general hospitals, for about 14 million people in the catchment area, which provide comprehensive eye care services including cataract surgical services in North West Ethiopia. However, all of them are urban-centered and not accessible for more than 80% of the community. Hence, the outreach program was planned for communities where cataract surgical services are not available and for peoples in disadvantaged areas with low socioeconomic status in North West Ethiopia. Initially, screening for cataracts was conducted in the nearby rural weredas, which are the third-level administrative units in the Ethiopian governance structure, before the campaign. During the screening, Cataract cases were diagnosed and informed about the condition and scheduled to attend the planned outreach program. As a result, a total of 1336 cataract cases were linked and admitted to the campaign from weredas of West Gojjam(628cases), South Gondar (374cases) Awi Zone (201 cases), Bahir Dar City (100 cases) and from East Gojjam (33cases). Thus more than 93% of cases were from rural weredas and district towns. Therefore, the findings of this study could apply to rural residents admitted for cataract surgery in outreach programs.”

Question 08

The ‘Study population and sampling technique’ section need to improve its clarity in describing the sampling technique used for the study. A simple random sampling technique requires selecting subjects from a predetermined sampling frame. If the list of 1336 cataract patients was used as the sampling frame, then at least two things need to be clarified.

1. The sampling frame was prepared based on the patients who visited the facility from June 10, 2018, to June 17, 2018; were the subjects chosen daily or based on an aggregated list.

Authors’ response

Dear reviewer thanks for the comments and questions.

It is rewritten on page 6 as “Initially, screening for cataracts was conducted in the nearby rural weredas, which are the third-level administrative units in the Ethiopian governance structure, before the campaign. During the screening, Cataract cases were diagnosed and informed about the condition and scheduled to attend the planned outreach program. As a result, a total of 1336 cataract cases were linked and admitted to the campaign from weredas of West Gojjam(628cases), South Gondar (374cases) Awi Zone (201 cases), Bahirdar City (100 cases) and from East Gojjam (33cases).

Thus more than 93% of cases were from rural weredas and district towns. Therefore, the findings of this study could apply to rural residents admitted for cataract surgery in outreach programs.’’

1. As the program provides free of cost cataract surgery, the subjects interviewed in this study should already have a preconceived estimation about the cost of the treatment; this has a definite impact on the elicited WTP (e.g. only 55 percent respondents elicited a non-zero WTP).

Authors’ response

Dear reviewer thanks for the comments and questions.

We have developed a case Scenario to minimize information asymmetry regarding cataract surgery and to estimate how beneficiaries value this service rationally. Even if we use the Case scenario to minimize strategic bias, the WTP approach is deemed not free of the afro-mentioned problem. So we have stated it in the limitation section. we have attached the case scenario at the supplementary file with the data collection tool. 

Case Scenario

” Introduction: Cataract is clouding of the eyes natural lenses and it is the leading cause of blindness worldwide. This problem can be caused by aging, family history, hypertension, obesity, diabetes mellitus, smoking, significant alcohol consumption, and other courses. To overcome this problem cataract surgery is the only treatment option. This outreach program which is designed for cataract surgery is funded by foreign donors and aims to fight blindness due to cataract by assuring accessibility and equity of the services for people living in districts. You are getting these services by now free of charge, but this is because of the cost is covered by funding organizations. However, in the future, this service will no longer be funded and sustainable.

Benefits of cataract surgery:

The cataract surgery helps to improve vision, prevent avoidable blindness, increasing productivity, and improve quality of life. As we have said before to make the service sustainable and accessible, designing a cost-recovery model is very important.

Question 09

The description of the bidding procedure given on page 5 needs clarity. How was the initial bid (i.e. 1000 ETB) was chosen? A schematic representation of the DBDC experiment would help the reader to visualize the context better

Authors’ response

Dear reviewer thanks for your comments and questions.

The biding experiment is put on page 9 now. It is clarified again as follows on page 8, as “The starting price for the biding game was estimated from the actual cost of manual small incision cataract surgery per eye, which was 2824 ETB from a study done in Southern Ethiopia in 2010 (34). In the Ethiopian context, the average household out pocket expenditure from the total health financing is about 35%. Based on this rate, the starting price that we expected to be paid was 1000 Birr (0.35x2824birr). During eliciting, the participants were asked their willingness to pay something for the Manual Small incision cataract surgery initially. If the participant was willing to pay, he/she was asked WTP for the target price (ETB 1000 per eye).’’

Question 10

About 45 percent of respondents were not willing to pay a positive price for the service, how were these responses handled in the analysis? Besides, if some people were not willing to pay, were these people statistically different from those that were willing to pay?

Authors’ response

Dear reviewer thanks for the comments and questions. 

To handle those participants who are not willing to accept to pay any price for the cataract surgery, Tobit regression econometric model was used. This model also considered as censored regression model used to estimate the parameters of how explanatory variables affect the latent dependent variable. So, 45 % of participants who are considered as censored for the lower limit were included in the model with an analysis of the Tobit regression model.

Question 11

The DBDC experiment ends with an interval of WTP. Did you check the interval regression techniques to estimate the associated factors? (If yes) Were the results different from an interval regression model?

Authors’ response

Dear reviewer thanks for the comments and questions. 

We are using the Contingent valuation method with a bidding game elicitation approach to explore the participants’ maximum willingness to pay for cataract surgery. The participants were asked as biding up or down based on their responses to the preceding price. The end question given for the participants was not Yes/No, rather an open question about their maximum willingness to pay. So, the final response for all participants who are willing to pay any positive price for cataract surgery was open continues price, not kind of Yes/No answer for the stated price. We only use those sated prices to elicit and manage the biding game to get the maximum value participants will give for the service. Due to this, we didn’t run the interval regression to identify associated factors.

Question 12

The average WTP can be estimated using a constant only interval regression model. Although, 45 percent respondents did not elicit a positive WTP, their true WTP might be greater than zero but less than the last bid (750 ETB). In that case counting them as zero (as done in this manuscript) might not be pragmatic

Authors’ response

Dear reviewer thanks for the comments and questions.

After presenting the scenario the initial question raised for the participants was as they willing to accept some price for the cataract surgery or not. So, those who were not willing any amount of payment for the cataract surgery (around 45% of the participants) waived the next biding game which was used to explore the maximum amount of price they want to pay for the service. The biding game approach was only applied to those participants (55%) who were willing or volunteer to pay some price for the service. Those participants who were not willing to pay any positive price were considered as a censored sample.

Question 13

The robustness of the Tobit models can be checked estimating coefficients of both censored and truncated samples as-well-as estimating the OLS coefficients. Please report the results (as supplementary documents). This practice will help the readers understand the robustness of the results obtained by the study.

Authors’ response

Dear reviewer thanks for the comments and questions. 

We have estimated the coefficients using the censored (Tobit) regression model, truncated regression model, and using the OLS model and attached as a supplementary file.

Question 14

The ‘Participant’s health-related characteristics’ section requires to provide sufficient details of the variables (what and how the measurements were taken; e.g. Systemic illness). 

Authors’ response 

Dear reviewer thanks for the comments and questions. 

The respondents were asked about their self-reported systemic illness.

Question 15

The discussion and conclusion need to be improved. 

Authors’ response

Dear reviewer thanks for your comments and questions. 

We tried to revise and rewrite the discussion and the conclusion.

Reviewer 02

Question 01

The overall standard of academic English is low and one has to struggle to read between the lines to understand the otherwise important information being conveyed. May I suggest the authors seek professional assistance is preparing the grammatical content of the manuscript.

Authors’ response

Dear reviewer thanks for your comments and questions. 

We have revised the manuscript both grammatically and content-wise.

Question 02

Many of the references are out of date. There are more recent references on the prevalence and causes of blindness, globally.

Authors’ response

Dear reviewer thanks for your comments and questions 

We have revised the references in the main document

Question 03

In the Methods section, the authors state that all patients admitted for cataract surgery at Tebbe Gion Hospital formed the study population, then mentions that the patients were screened for cataract. What was the purpose of the screening? When was this study conducted?

Authors’ response

Dear reviewer thanks for your comments and questions. 

 Dear reviewer thanks for the comments. We have put detailed information on how and why cataract pre-screening was done and when was the study conducted on pages 6 and 7. We put it as follows. 

“An Outreach- based cross-sectional study was conducted. The campaign site was at Tebebe Gion Specialized Referral Hospital, Bahirdar city, North West Ethiopia. There are only four Tertiary and general hospitals, for about 14 million people in the catchment area, which provide comprehensive eye care services including cataract surgical services in North West Ethiopia. However, all of them are urban-centered and not accessible for more than 80% of the community. Hence, the outreach program was planned for communities where cataract surgical services are not available and for peoples in disadvantaged areas with low socioeconomic status in North West Ethiopia. Initially, screening for cataracts was conducted in the nearby rural weredas, which are the third-level administrative units in the Ethiopian governance structure, before the campaign. During the screening, Cataract cases were diagnosed and informed about the condition and scheduled to attend the planned outreach program. As a result, a total of 1336 cataract cases were linked and admitted to the campaign from weredas of West Gojjam(628cases), South Gondar (374cases) Awi Zone (201 cases) Bahirdar (100 cases) and from East Gojjam (33cases). Thus more than 93% of cases were from rural weredas and district towns. Therefore, the findings of this study could apply to rural residents admitted for cataract surgery in outreach programs.’’

Question 04

Several unfamiliar analytical terms are mentioned. 

Double Bounded Dichotomous Choice Variant of the Contingent Valuation (CV) 

Principal Component Analysis (PCA)

Homoscedasticity of error terms through Breush-pagan for Heterosekedasticity test

The authors should briefly explain what they are and put a reference as this is a broad remit medical journal

Authors’ response

Dear reviewer thanks for your comments and questions.

We have explained these terms in the main document method part and we have also included the reference.

Question 05

We didn’t measure visual acuity rather Visual ability which is the overall visual performance including visual acuity, color vison, contrast sensitivity, and visual field. So, the visual ability was measured using modified VF-14 tool, not Snellen’s notation or the log MAR.

Reviewer 03

Question 01

I found it quite difficult to understand the text because English is not written in standard syntax and grammar. I strongly suggest that the authors engage the service of a professional English editing service before re-review of the manuscript

Authors’ response

Dear reviewer thanks for your comments and questions

We have revised the manuscript both content-wise and grammatically.

Question 02

Besides, the methods and statistics require some more explanation for clarity and ease of understanding by the reader.

Authors’ response

Dear reviewer thanks for your comments and questions. 

We explain those terms like WTP, CV method, Bidding game elicitation approach, and Tobit regression econometric model at the method section.

---

## [Decision Letter · Decision Letter 1]

21 Aug 2020

PONE-D-19-31270R1

Willingness to Pay for Cataract Surgery and Associated Factors among Cataract Patients in Outreach Site, North West Ethiopia

PLOS ONE

Dear Dr. Amare,

Thank you for submitting your manuscript to PLOS ONE. After careful consideration, we feel that it has merit but does not fully meet PLOS ONE’s publication criteria as it currently stands. Therefore, we invite you to submit a revised version of the manuscript that addresses the points raised during the review process.

In revising your manuscript, please consider all the comments from the reviewers carefully and elaborate on how these were addressed individually in your response to reviewers. In general, reviewers see value in your work. However, they have highlighted a number of residual and additional concerns on your manuscript. There are also several errors in the quality of grammar and writing. I urge you to consult a native English speaker or professional services to improve on the content. Your manuscript can be considered for further review only if you are willing to address these in your next revision.

We look forward to receiving your revised manuscript.

Kind regards,

Janhavi Ajit Vaingankar

Academic Editor

PLOS ONE

Reviewers' comments:

Reviewer's Responses to Questions

**Comments to the Author**

1. If the authors have adequately addressed your comments raised in a previous round of review and you feel that this manuscript is now acceptable for publication, you may indicate that here to bypass the “Comments to the Author” section, enter your conflict of interest statement in the “Confidential to Editor” section, and submit your "Accept" recommendation.

Reviewer #1: (No Response)

Reviewer #2: (No Response)

Reviewer #3: (No Response)

2. Is the manuscript technically sound, and do the data support the conclusions?

Reviewer #1: Yes

Reviewer #2: Partly

Reviewer #3: Partly

3. Has the statistical analysis been performed appropriately and rigorously? 

Reviewer #1: Yes

Reviewer #2: I Don't Know

Reviewer #3: I Don't Know

4. Have the authors made all data underlying the findings in their manuscript fully available?

Reviewer #1: No

Reviewer #2: Yes

Reviewer #3: Yes

5. Is the manuscript presented in an intelligible fashion and written in standard English?

Reviewer #1: No

Reviewer #2: No

Reviewer #3: No

6. Review Comments to the Author

Reviewer #1: [PONE-D-19-31270R1]

Title: Willingness to pay for cataract surgery and associated factors among cataract patients in outreach site, North West Ethiopia

I thank the editor for the opportunity to review the manuscript once again. I also appreciate the authors’ for the efforts given to revise the manuscript according to the suggestions shared. However, there are a few important issues that require additional revisions or clarifications. The suggestions are as follows:

In reply to the authors’ response to the Reviewer #1-Comment #1:

The schematic description can provide a summarized picture of the bidding, however, it will not help the reader to replicate the experiment precisely in a similar manner. Including the tool as a supplementary document will provide the chance to the reader to understand what information was given to the respondents before and while eliciting WTP. Also, it would help the reader to design a similar questionnaire/tool for their experiments.

In reply to the authors’ response to the Reviewer #1-Comment #5:

In the previous review, the author was suggested to shed some light on the health-state dependency of the WTP and discuss how that would affect the results of their study. Looking at the available literature in this context would have been an appreciable initiative. However, the current version of the manuscript neither properly discussed nor provided any reference on their statements made in this context. There are researches that provide theoretical background as-well-as empirical evidences that WTP is health-state dependent. I would request the author to re-address comment #5.

In reply to the authors’ response to the Reviewer #1-Comment #9:

The schematic description of the DBDC experiment (shown on page 9) does not contain the percentage of respondents on each of the decision nodes and the interval they ended up before answering the maximum WTP and the average maximum WTP for different paths taken. It would provide the reader with a summary of the bidding experiment and would help understand whether the prices used for the experiment were reasonable or there were too many censored observations at any particular price points. I would strongly recommend the authors consider adding the information on the diagram shown on page 9.

Moreover, the statement made in lines 147-148 of page 8 needs a proper reference. Can you please add any literature that provides a background of this 35 percent out of pocket health expenditure as a proportion of the total healthcare spending of Ethiopia?

In reply to the authors’ response to the Reviewer #1-Comment #12:

The authors mentioned that 45 percent of the respondents who did not elicit any positive WTP were excluded from the further bidding experiments. There are three issues that need to be addressed:

1. The current schematic description (shown on page 9) does not include the initial question regarding the positive WTP. Please revise it accordingly.

2. If 45 percent of the respondents are not willing to pay any positive price, then, are there any systematic differences in their sociodemographic and/or health-related characteristics? Please provide both pooled and separated descriptive statistics in Table 1 and 2. Also, test the statistical significance of the differences by conducting either bivariate analyses or test a multivariable selection model for the two groups (i.e. =1 if positive WTP, 0 otherwise).

3. As these 45 percent of the respondents did not participate in the second phase of the bidding game, the Tobit model should also be estimated for both pooled and samples with only positive WTP. The authors have already estimated the models while addressing Comment #13. However, authors may consider showing and discussing them with the main results instead of just keeping in the appendix.

General suggestions:

The regression table shared as the supplementary document does not contain all the necessary information required to assess the robustness of the results. Please provide at least the standard errors and indications of statistical significance for all the models estimated.

There are a few minor grammatical errors and typos that require correction (e.g. see lines 21-23 on page 2). There are a few technical errors too that demand authors’ attention (e.g. lines 28-29 on page 2; simple random sampling).

If possible, please use an equation writing tool (such as equation writer in MS Word) and convert the lines 230-231 into an equation.

The data needs additional clarifications and missing the original variables. Standard practice should be to share the cleaned data according to the questionnaire/tools shared. Moreover, the MWTP data contains decimal characters; are these original MWTP or convert figures?

Reviewer #2: This is a very important topic and the authors should be congratulated on carrying out this research. However, there are several concerns that have not been addressed.

Technical Issues

1. Many of the references are out of date. There is more recent data on the prevalence and causes of global blindness and visual impairment. Flaxman et al 2017, Bourne et al 2017

2. Several statements in the introduction are non- factual.

E.g. Cataract accounts for a third of all global blindness, not 51% as you stated.

Universal Eye Health from the World Health Organization's (WHO) Global Action Plan for eye health was launched in 2013, yet you provide references from 2001 and 2011.

3.. Visual acuity as classified by the WHO International classification of disease is the most common descriptor for assessing vision. You insist on using visual ability. The references you quote for visual ability are silent on this terminology.16, 18, 20-22, 24-27

Grammatical issues.

4. Incorrect grammar and syntax through out the article.

Reviewer #3: I appreciate the fact that the authors have made concerted efforts to improve the quality of the manuscript. However, I still had considerable difficulty with understanding the text and deciphering the message that the authors want to communicate to the reader. I do understand that English is not the first language of the authors but in order to ensure effective communication, the manuscript must be intelligible. Therefore, I wish to repeat my suggestion that the authors engage the service of a professional English editing service before re-review of the manuscript.

7. PLOS authors have the option to publish the peer review history of their article (what does this mean?). If published, this will include your full peer review and any attached files.

Reviewer #1: No

Reviewer #2: **Yes: **Ada Aghaji

Reviewer #3: No

---

## [Author Response · Author response to Decision Letter 1]

28 Sep 2020

Response reviewers

Reviewer 1

Issues raised by the reviewer on comment #1

Authors’ response 

Dear reviewer, thanks for your comments and questions. 

We have revised the manuscript as per the comment, and we have added the bidding approach tool we used as a supplementary file.

Issues raised by the reviewer on comment #5

Authors’ response 

Dear reviewer, thanks for your comments and questions. 

We have revised the main manuscript on page 5 from line 75 to 78. We have included the health state dependant nature of WTP.

Issues raised by the reviewer on comment #9

Dear reviewer, thanks for your comments and questions.

We have revised the main manuscript as per the given comment. We have analyzed the percentage of respondents at each node and stated in the main document on page 9. 

We have also included the reference in the main manuscript for the issue raised on page 8.

Issues raised by the reviewer on comment #12

1. The current schematic description (shown on page 9) does not include the initial question regarding the positive WTP. Please revise it accordingly.

Authors’ response 

Dear reviewer, thanks for your comments and questions. 

We have revised the main manuscript accordingly, as per the comment on page 8.

2. If 45 percent of the respondents are not willing to pay any positive price, then are there any systematic differences in their sociodemographic and/or health-related characteristics? Please provide both pooled and separated descriptive statistics in Table 1 and 2. Also, test the statistical significance of the differences by conducting either bivariate analyses or test a multivariable selection model for the two groups (i.e. =1 if positive WTP, 0 otherwise).

Authors’ response 

Dear reviewer, thanks for your comments and questions. 

We have revised the main manuscript, and we present the pooled and separated descriptive statistics on table 1 and table 2. We have also run the logistic regression for WTP (1 = willing to pay and 0 = otherwise), and we have presented the finding in the main manuscript as table 3.

3. As these 45 percent of the respondents did not participate in the second phase of the bidding game, the Tobit model should also be estimated for both pooled and samples with only positive WTP. The authors have already estimated the models while addressing Comment #13. However, authors may consider showing and discussing them with the main results instead of just keeping in the appendix.

Authors’ response 

Dear reviewer, thanks for your comments and questions.

We have revised the table and included in the main manuscript as table 5.

For 1eneral suggestions raised by the reviewer 

1. The regression table shared as the supplementary document does not contain all the necessary information required to assess the robustness of the results. Please provide at least the standard errors and indications of statistical significance for all the models estimated.

Authors’ response 

Dear reviewer, thanks for your comments and questions.

We have provided the standard error and p-value for the regression table and presented as table 5 in the main manuscript.

2. There are a few minor grammatical errors and typos that require correction (e.g., see lines 21-23 on page 2). There are a few technical errors that demand authors’ attention (e.g., lines 28-29 on page 2; simple random sampling). 

Authors’ response 

Dear reviewer, thanks for your comments and questions.

We have revised the manuscript as per the comment.

3. If possible, please use an equation writing tool (such as equation writer in MS Word) and convert the lines 230-231 into an equation.

Authors’ response 

Dear reviewer, thanks for your comments and questions.

We have revised it and used the equation writing tool

4. The data needs additional clarifications and missing the original variables. Standard practice should be to share the cleaned data according to the questionnaire/tools shared. Moreover, the MWTP data contains decimal characters; are this original MWTP or convert figures?

Authors’ response 

Dear reviewer, thanks for your comments and questions.

The MWTP is the converted one since initially, the MWTP was collected in Ethiopian Birr (ETB) and converted to USD for reporting and analysis. We have included the variables used for analysis in the main data, and the wealth index variable is computed from data collected from household assets by PCA.

Reviewer 2

Question 1

1. Many of the references are out of date. There are more recent data on the prevalence and causes of global blindness and visual impairment. Flaxman et al 2017, Bourne et al 2017

Authors’ response 

Dear reviewer, thanks for your comments and questions.

We have revised the main manuscript as per the given comment.

2. Several statements in the introduction are non- factual.

E.g., Cataract accounts for a third of all global blindness, not 51%, as you stated.

Universal Eye Health from the World Health Organization’s (WHO) Global Action Plan for eye health was launched in 2013, yet you provide references from 2001 and 2011.

Authors’ response 

Dear reviewer, thanks for your comments and questions.

We have revised the main manuscript as per the given comment on page 4, lines 49 to 51.

3. Visual acuity, as classified by the WHO International classification of disease, is the most common descriptor for assessing vision. You insist on using a visual ability. The references you quote for visual ability are silent on this terminology.16, 18, 20-22, 24-27

Authors’ response 

Dear reviewer, thanks for your comments and questions.

We were considering to measure overall visual function with visual ability, which includes visual acuity as one component. As you stated, most literature used visual acuity to measure visual function. Visual acuity and visual ability have scope differences in which visual acuity is more specific, and visual ability is a general one. We want to acknowledge the comment you give us, and we appreciate the difference between visual ability and visual acuity. We use the references used for visual acuity because of both visual acuity, and visual ability reflects the visual function of the participants. We admit the comment you gave us, and if necessary, we can mention this in the limitation section.

4. Incorrect grammar and syntax throughout the article.

Authors’ response 

Dear reviewer, thanks for your comments.

We tried to revise and edit the grammar of the main manuscript accordingly.

Reviewer 3

Incorrect grammar and language 

Authors’ response 

Dear reviewer, thanks for your comments.

We tried to revise and edit the grammar of the main manuscript accordingly.

---

## [Decision Letter · Decision Letter 2]

11 Nov 2020

PONE-D-19-31270R2

Willingness to Pay for Cataract Surgery and Associated Factors among Cataract Patients in Outreach Site, North West Ethiopia

PLOS ONE

Dear Dr. Amare,

Thank you for submitting your manuscript to PLOS ONE. After careful consideration, we feel that it has merit but does not fully meet PLOS ONE’s publication criteria as it currently stands. Therefore, we invite you to submit a revised version of the manuscript that addresses the points raised during the review process.

We look forward to receiving your revised manuscript.

Kind regards,

Janhavi Ajit Vaingankar

Academic Editor

PLOS ONE

Reviewers' comments:

Reviewer's Responses to Questions

**Comments to the Author**

1. If the authors have adequately addressed your comments raised in a previous round of review and you feel that this manuscript is now acceptable for publication, you may indicate that here to bypass the “Comments to the Author” section, enter your conflict of interest statement in the “Confidential to Editor” section, and submit your "Accept" recommendation.

Reviewer #1: (No Response)

Reviewer #2: (No Response)

2. Is the manuscript technically sound, and do the data support the conclusions?

Reviewer #1: Partly

Reviewer #2: No

3. Has the statistical analysis been performed appropriately and rigorously? 

Reviewer #1: Yes

Reviewer #2: I Don't Know

4. Have the authors made all data underlying the findings in their manuscript fully available?

Reviewer #1: No

Reviewer #2: Yes

5. Is the manuscript presented in an intelligible fashion and written in standard English?

Reviewer #1: No

Reviewer #2: Yes

6. Review Comments to the Author

Reviewer #1: I thank the editor for the opportunity to review the manuscript for the third time. I also appreciate the authors’ for the efforts given to revise the manuscript according to the suggestions shared. However, there are a few inconsistencies that need to be corrected. My comments are as follows:

1. Please mention the dates of data collection also in the main body.

2. The exchange rate used for the survey is the one of 2018. However, the world bank data the manuscript refers to in line no. 149 on page 8 is of the year 2009 (see reference no. 43). Moreover, the manuscript mentions that the out of pocket health expenditure is 35 percent of the total health expenditure in Ethiopia. Whereas, the report it refers to in reference no. 43 shows that it is 37 percent for Ethiopia. Authors should be very transparent and careful about the methodologies used and the methodologies described in the manuscript.

3. Line no. 152-154 mentions that the respondents were asked – “Are you willing to pay some positive price for cataract surgery?”, however, the questionnaire attached as a supplementary file states that, “by considering the above scenario, if the service is vailing for you, are you willing to pay something for this service”. Although, the sentences provide the same understanding of the question, there should not be any inconsistencies between what is written in the manuscript and what was actually asked. If the author wants to describe this as part of the methodology they applied; then, the exact question need not to be mentioned. This could be written differently, such as, “After briefing the case scenario, all the participants were asked whether they are willing to pay any positive price for the proposed cataract surgery”.

4. The intervals are not correctly specified in the schematic diagram. If anyone is not willing to pay 500 ETB, then the interval they fall into is 0 to <500 ETB. They intervals should be fixed by the experiment-design, not the average maximum WTP.

5. The dataset still does not contain the bidding experiment data. Sharing partial dataset will not help serving the right purpose of data-sharing.

Reviewer #2: The authors have addressed some of my comments. The main concern I had and still have is about the visual ability measurement. This is not a regular term and the authors have not defined it or explained how it was assessed. The crux of the paper is the relationship between visual ability and WTP, so this is a major flaw which the authors have not addressed. Visual acuity is usually used for WTP research. The references provided about visual ability refer to visual acuity.

7. PLOS authors have the option to publish the peer review history of their article (what does this mean?). If published, this will include your full peer review and any attached files.

Reviewer #1: No

Reviewer #2: No

---

## [Author Response · Author response to Decision Letter 2]

5 Dec 2020

Reviewer 1 comments and questions: -

1. Please mention the dates of data collection also in the main body.

Author response

Dear reviewer, thanks for your comments and questions

We have revised the manuscript as per the given comment. Please see page 7 and line 136 of the revised manuscript.

2. The exchange rate used for the survey is the one of 2018. However, the world bank data the manuscript refers to in line no. 149 on page 8 is of the year 2009 (see reference no. 43). Moreover, the manuscript mentions that the out of pocket health expenditure is 35 percent of the total health expenditure in Ethiopia. Whereas, the report it refers to in reference no. 43 shows that it is 37 percent for Ethiopia. Authors should be very transparent and careful about the methodologies used and the methodologies described in the manuscript.

Author response

Dear reviewer, thanks for your comments and questions

We have revised the reference, and we have removed the previous reference, and we have added the world bank group 2017 report of Ethiopia and Sub-Saharan African countries. Please see page 8 and lines 148 to 150 of the revised manuscript.

3. Line no. 152-154 mentions that the respondents were asked – “Are you willing to pay some positive price for cataract surgery?”, however, the questionnaire attached as a supplementary file states that, “by considering the above scenario, if the service is vailing for you, are you willing to pay something for this service”. Although, the sentences provide the same understanding of the question, there should not be any inconsistencies between what is written in the manuscript and what was actually asked. If the author wants to describe this as part of the methodology they applied; then, the exact question need not to be mentioned. This could be written differently, such as, “After briefing the case scenario, all the participants were asked whether they are willing to pay any positive price for the proposed cataract surgery”.

Author response 

Dear reviewer, thanks for your comments and questions

We have reviewed it, and we have made it consistent have was written on the methodology and the tool part.

4. The intervals are not correctly specified in the schematic diagram. If anyone is not willing to pay 500 ETB, then the interval they fall into is 0 to <500 ETB. The intervals should be fixed by the experiment-design, not the average maximum WTP.

Author response 

Dear reviewer, thanks for your comments and questions. 

We tried to revise the intervals in each experimental design. Please see page 9 figure of the revised manuscript. 

5. The dataset still does not contain the bidding experiment data. Sharing partial dataset will not help serving the right purpose of data-sharing.

Author response

Dear reviewer, thanks for your comments and questions

We have included the biding experiment data in the revised data set. Please see the revised data set.

Reviewer 2 comments and questions: -

1. The authors have addressed some of my comments. The main concern I had and still have is about the visual ability measurement. This is not a regular term and the authors have not defined it or explained how it was assessed. The crux of the paper is the relationship between visual ability and WTP, so this is a major flaw which the authors have not addressed. Visual acuity is usually used for WTP research. The references provided about visual ability refer to visual acuity.

Authors response 

Dear reviewer, thanks for your comments and questions

We have accepted your comments, and we measured pre-operative visual ability by using the Visual Function 14 Index (VF-14) tool. The response option was prepared with Likert’s scale format. The scale ranged from the 0-4 value for the degree of visual difficulties ‘unable to do’ (scale 0), ‘great difficulty’ (scale1), ‘moderate difficulty’ (scale 2), ‘little difficulty’ (scale 3) and ‘no difficulty’ (scale 4). Then each scale is multiplied by 25. So, the value of each response ranges from 0-100. After that, the factored amounts were summed up. Finally, VF index was computed by dividing the summed factored amounts to the number of checked boxes. Visual ability was categorized as normal or near-normal performance (≥50 visual ability score) and restricted performance (<50 visual ability score) based on the international council of Ophthalmology visual ability classification. We have included this in the method section on page 10 and lines 180 to 189 of the revised manuscript.

---

## [Decision Letter · Decision Letter 3]

6 Jan 2021

PONE-D-19-31270R3

Willingness to Pay for Cataract Surgery and Associated Factors among Cataract Patients in Outreach Site, North West Ethiopia

PLOS ONE

Dear Dr. Amare,

Thank you for submitting your manuscript to PLOS ONE. After careful consideration, we feel that it has merit but does not fully meet PLOS ONE’s publication criteria as it currently stands. Therefore, we invite you to submit a revised version of the manuscript that addresses the points raised during the review process.

We look forward to receiving your revised manuscript.

Kind regards,

Janhavi Ajit Vaingankar

Academic Editor

PLOS ONE

Reviewers' comments:

Reviewer's Responses to Questions

**Comments to the Author**

1. If the authors have adequately addressed your comments raised in a previous round of review and you feel that this manuscript is now acceptable for publication, you may indicate that here to bypass the “Comments to the Author” section, enter your conflict of interest statement in the “Confidential to Editor” section, and submit your "Accept" recommendation.

Reviewer #1: All comments have been addressed

Reviewer #2: (No Response)

2. Is the manuscript technically sound, and do the data support the conclusions?

Reviewer #1: Yes

Reviewer #2: Partly

3. Has the statistical analysis been performed appropriately and rigorously? 

Reviewer #1: Yes

Reviewer #2: I Don't Know

4. Have the authors made all data underlying the findings in their manuscript fully available?

Reviewer #1: Yes

Reviewer #2: Yes

5. Is the manuscript presented in an intelligible fashion and written in standard English?

Reviewer #1: Yes

Reviewer #2: No

6. Review Comments to the Author

Reviewer #1: Thank you for considering the suggestions and making all the efforts in revising accordingly. I also thank the editor for sharing such an interesting work. Best wishes.

Reviewer #2: Some of my previous comments have been addressed. However, there are still a few concerns.

1. The quality of the grammar and syntax is not of academic standard and may require the services of a professional to bring it up to the required standard.

2. The authors have explained their methods for calculating visual ability. However, they should include references for the calculation and examples of where this has been used in ophthalmic literature. Visual ability is not a term commonly used in ophthalmic literature and the thesis of this manuscript is the relationship between visual ability and willingness to pay.

3. There are a few statements that are not factually correct. The opening sentence states that cataract is the second commonest cause of blindness worldwide; Cataract is the commonest cause of blindness globally. Refractive error is the commonest cause of visual impairment globally.

4. Cataract surgical coverage and Cataract Surgical rate are two different indicators for assessing the quantity of cataract surgery.

5. In sum therefore, I believe that this article has some merit, but a lot of work still needs to be done.

7. PLOS authors have the option to publish the peer review history of their article (what does this mean?). If published, this will include your full peer review and any attached files.

Reviewer #1: No

Reviewer #2: **Yes: **Ada Aghaji

---

## [Author Response · Author response to Decision Letter 3]

3 Feb 2021

Response to Reviewers

Reviewer 2 comments and questions: -

Some of my previous comments have been addressed. However, there are still a few concerns.

1. The quality of the grammar and syntax is not of academic standard and may require the services of a professional to bring it up to the required standard.

Author Response

Dear reviewer thank you for your comments and questions. We have revised the English language of the manuscript by the professional language editor. Please see the whole revise manuscript. 

2. The authors have explained their methods for calculating visual ability. However, they should include references for the calculation and examples of where this has been used in ophthalmic literature. Visual ability is not a term commonly used in ophthalmic literature and the thesis of this manuscript is the relationship between visual ability and willingness to pay.

Author Response

Dear reviewer thank you for your comments and questions. The function of the visual system can be measured either objectively or subjectively. Visual acuity measurement is one of the objective indicators of the visual system functionality. However, it does not indicate the overall visual performance of an individual. Hence, we used the subjective measurement, the visual function/visual ability, to assess the overall visual performance. Hence it was measured by using adapted Visual Function 14 Index (VF-14) tool. Previously, numerous researchers used the tool as a standard. However, based on the socio-economic and cultural perspective of their study population, the tool was adapted. The response option was prepared with 5-point Likert’s scale format. The scale ranged from the 0-4 value for the degree of visual difficulties ‘unable to do’ (scale 0), ‘great difficulty’ (scale1), ‘moderate difficulty’ (scale 2), ‘little difficulty’ (scale 3) and ‘no difficulty’ (scale 4). Then each scale was multiplied by 25. So, the value of each response ranges from 0-100. After that, the factored amounts were summed up. Finally, VF index was computed by dividing the summed factored amounts to the number of checked boxes and the visual ability were categorized. 

The calculation is elaborated across the studies, and the final classification is based on the International Council of Ophthalmology as normal or near-normal performance(≥50 visual ability score) and restricted performance(<50 visual ability score). We have included references in the revised manuscript at page 

3. There are a few statements that are not factually correct. The opening sentence states that cataract is the second commonest cause of blindness worldwide; Cataract is the commonest cause of blindness globally. Refractive error is the commonest cause of visual impairment globally.

Author Response

Dear reviewer thank you for your comments and questions 

4. Cataract surgical coverage and Cataract Surgical rate are two different indicators for assessing the quantity of cataract surgery

Author Response

Dear reviewer thank you for your comments and questions. We have revised the manuscript based o the raised issue. Please see page 4 from lines 51 to 53 of the revised manuscript.

---

## [Decision Letter · Decision Letter 4]

3 Mar 2021

Willingness to Pay for Cataract Surgery and Associated Factors among Cataract Patients in Outreach Site, North West Ethiopia

PONE-D-19-31270R4

Dear Dr. Amare,

We’re pleased to inform you that your manuscript has been judged scientifically suitable for publication and will be formally accepted for publication once it meets all outstanding technical requirements.

Kind regards,

Janhavi Ajit Vaingankar

Academic Editor

PLOS ONE

Additional Editor Comments (optional):

Reviewers' comments:

Reviewer's Responses to Questions

**Comments to the Author**

1. If the authors have adequately addressed your comments raised in a previous round of review and you feel that this manuscript is now acceptable for publication, you may indicate that here to bypass the “Comments to the Author” section, enter your conflict of interest statement in the “Confidential to Editor” section, and submit your "Accept" recommendation.

Reviewer #1: All comments have been addressed

2. Is the manuscript technically sound, and do the data support the conclusions?

Reviewer #1: Yes

3. Has the statistical analysis been performed appropriately and rigorously? 

Reviewer #1: Yes

4. Have the authors made all data underlying the findings in their manuscript fully available?

Reviewer #1: Yes

5. Is the manuscript presented in an intelligible fashion and written in standard English?

Reviewer #1: Yes

6. Review Comments to the Author

Reviewer #1: Thank you for considering the suggestions and making all the efforts in revising accordingly. I also thank the editor for sharing such an interesting work. Best wishes.

7. PLOS authors have the option to publish the peer review history of their article (what does this mean?). If published, this will include your full peer review and any attached files.

Reviewer #1: No

---

## [Editor Report · Acceptance letter]

11 Mar 2021

PONE-D-19-31270R4 

Willingness to Pay for Cataract Surgery and Associated Factors among Cataract Patients in Outreach Site, North West Ethiopia 

Dear Dr. Amare:

I'm pleased to inform you that your manuscript has been deemed suitable for publication in PLOS ONE. Congratulations! Your manuscript is now with our production department. 

Kind regards, 

on behalf of

Ms Janhavi Ajit Vaingankar 

Academic Editor

PLOS ONE